# Exploring Cervical Cancer Screening Uptake among Women in the United States: Impact of Social Determinants of Health and Psychosocial Determinants

**DOI:** 10.3390/bs14090811

**Published:** 2024-09-13

**Authors:** Matthew Asare, Eyram Owusu-Sekyere, Anjelica Elizondo, Gabriel A. Benavidez

**Affiliations:** Department of Public Health, Robbins College of Health and Human Sciences, Baylor University, Waco, TX 76798, USA; eyram_owususekyere1@baylor.edu (E.O.-S.); anjelica_elizondo@baylor.edu (A.E.); gabriel_benavidez@baylor.edu (G.A.B.)

**Keywords:** cervical cancer, social determinants of health, screening, psychosocial factors

## Abstract

Several non-medical factors, such as income, education, and access to care, directly or indirectly affect adherence to cancer screening guidelines. We examined the impact of social determinants of health (SDOH) and psychosocial factors on screening behavior in a nationally representative sample of women in the US. A retrospective population-level cross-sectional sample was extracted from the 2022 Health Information National Trends Survey. The dependent variables were the interest in cervical cancer screening and the screening behavior. The independent variables included SDOH and psychosocial factors. Descriptive statistics were calculated for demographics and covariates, and population-based estimates with 95% confidence intervals (CI) were produced for Pap testing behaviors. Logistic regression models assessed differences in Pap testing based on SDOH and psychosocial factors, adjusting for covariates. The study included 2224 women with a mean age of 46.96. Results showed that 90% of women were interested in cervical cancer screening, with an 80% screening rate. Screening rates varied by age and rurality. SDOH and psychosocial factors influenced both interest and actual screening, with 3% and 1% impacts, respectively. These findings suggest that SDOH and psychosocial factors are associated with cervical cancer screening uptake, highlighting the need for policies to address these disparities. Policies must be directed at bridging the gap created by these SDOHs. Public health professionals and researchers can design interventions using the SDOH and psychosocial frameworks to increase cervical cancer screening uptake.

## 1. Introduction

Gynecologic cancers have persisted as a significant cause of mortality among the female population, accounting for 30% of deaths among females with cancer across the globe [1]. Notable gynecologic cancers include uterine, ovarian, breast, and cervical cancers [1,2,3]. It is projected that approximately 13,820 women in the United States will be diagnosed with late-stage cervical cancer, and 4360 women will have succumbed to the disease by the end of 2024 [4]. Factors such as race, ethnicity, socioeconomic status, rurality, and religious disposition [5,6,7] are responsible for many of the disparities in incidence, stage at diagnosis, and consequent survival of cervical cancer in the United States. For example, non-Hispanic white women are diagnosed at an earlier stage of cervical cancer at diagnosis compared to black women [8]. Women with low incomes are more likely to be diagnosed with late-stage cervical cancer and have a higher mortality rate compared to women with higher incomes [4]. Rurality impacts the stage of cervical cancer diagnosis and survival; black women living in non-metropolitan areas have a lower survival rate compared to those living in metropolitan areas. Regardless of race, women living in non-metropolitan areas have higher odds of developing cervical cancer [9]. These disparities can be bridged through timely cervical cancer screening, as approximately 70% of deaths from cervical cancer can be prevented through routine screening [10].

The survival rate of cervical cancer is heavily dependent on stage at diagnosis, with approximately a 90% five-year survival rate when detected at stage 1 and only 10% when detected at stage 3 [7]. Unfortunately, one of the primary reasons for late-stage diagnoses is a lack of early detection through screening [11]. Women who receive regular screenings are less likely to be diagnosed at advanced stages of cervical cancer [11,12]. Among women between the ages of 35 and 64 years, cervical cancer screening has the potential to reduce the risk of late-stage diagnosis by 95% [13], underscoring the importance of regular screening for age-eligible women in the United States.

Screening tests, including HPV tests, Pap tests, and visual inspection with acetic acid (VIA), are used for the early detection of cervical cancer. The US Preventive Services Task Force (USPSTF), the American Cancer Society (ACS), and the American College of Obstetrics and Gynecology (ACOG) provide various recommendations for cervical cancer screening. While there are variations in screening guidelines across the agencies, there is a near consensus that women aged 21 to 29 should receive a Pap test every three years, and women aged 30 to 65 should receive either a Pap test alone every three years, an HPV test alone every five years, or both Pap and HPV tests together every five years [14]. Currently, the standard care for most Pap tests requires a physician to obtain samples from the cervix for further examination, while HPV tests require samples from the cervix that can be obtained using brushes or swabs, or other devices either by physicians or by screening participants. Consistent pap testing has reduced cervical cancer cases and mortality by at least 80% [15]. However, as of 2021, about 72.4% of age-eligible adult US women were currently on cervical cancer screening, over 11% short of the Healthy People 2030 targeted goal of 84.3% [15]. Additionally, an estimated 14 million eligible women in the US have not been screened [16], and women who have never been screened have a 50% greater risk of developing invasive cervical cancer [17]. 

While studies have found that the healthcare providers’ characteristics have an influence on uptake [18,19,20], there are several studies that have identified specific barriers to cervical cancer screening. These barriers include cost, fear of finding cancer, anxiety, embarrassment, the anticipation of pain, reluctance to be seen by a male provider, lack of knowledge about screening, language barriers, other health issues, lack of transportation, forgetting to schedule appointments, and lack of time [21,22,23,24,25,26]. 

Social determinants of health (SDOH) behavior and psychosocial factors contribute to low screening rates [27,28,29,30,31,32]. The social determinants of health (SDOH) behavior posit that non-medical factors, i.e., the conditions in which individuals are born, grow, live, work, and age, can directly or indirectly influence health behavior and outcomes [33]. SDOH behavioral factors include economic stability, neighborhood and built environment, education access and quality, community and social context, and the healthcare system [22,23]. A study assessing disparities in cancer screening uncovered that women who earned less than USD 50,000 annually had a lower prevalence of cervical cancer screening compared to those who earned more than USD 50,000 [34]. 

Similarly, evidence shows that psychosocial determinants, which include combinations of constructs from various behavioral theories, are associated with women’s cervical cancer screening behaviors. For instance, health belief models (e.g., perceived susceptibility to getting cancer, perceived severity [fatalism of cancer], and perceived beliefs that everything causes cancer) and the theory of planned behavior constructs (e.g., perceived behavioral control over the chances of getting cancer and the attitude toward recommendations for preventive measures [e.g., screening]) around the causes and treatment of cervical cancer affect women’s screening behaviors [27,28,29,30].

While previous studies have contributed to our understanding of the direct relationships between SDOH, psychosocial factors, and cervical cancer screening, those studies were limited in scope regarding the number of SDOH constructs applied in those studies [31,32,34]. For instance, studies that assessed the associations between SDOH and cervical cancer screening used either education and income, or only income, as proxies or a proxy of SDOH [31,32,34]. Other studies examined the impact of the rurality, immigration status, and ethnicity constructs of SDOH on screening behavior [35,36]. The opportunity exists to comprehensively operationalize SDOH subscales and psychosocial factors, and to determine the combined effect of both variables (i.e., SDOH and psychosocial ones) on cervical cancer screening. 

The purpose of this study was to examine which of the SDOH subscales and psychosocial determinants were significantly associated with cervical cancer screening behavior in a nationally representative sample of women in the US. We also assessed the cumulative impact of the overall SDOH and psychosocial determinants on participants’ screening behaviors.

## 2. Materials and Methods

### 2.1. Population

A retrospective population-level cross-sectional sample was extracted from the 2022 Health Information National Trends Survey (HINTS). Each year, the National Cancer Institute (NCI) administers the HINTS survey among nationally representative non-institutionalized adult populations aged 18 or older in the United States. The 2022 dataset was used because that was the only dataset that collected data on Americans’ health-related behaviors, comprehensive data on SDOH and psychosocial variables, and participants’ interest in screening and their cancer screening (Pap test) behaviors [37]. 

### 2.2. Measures

#### 2.2.1. Outcome Variables

The outcome variables included women’s (a) interest in cancer screening and (b) cervical cancer screening behavior, defined as a self-reported receipt of a Pap test. The outcome items were (1) “How interested are you in having a cancer screening test in the next year?” The responses were “not all”, “little”, “somewhat”, and “very”. We recorded the responses where “somewhat” and “very” were record as “Yes, interested” (1), and “not all” and “little” were recorded as “Not interested” (0). (2) “How long ago did you have your most recent Pap test to check for cervical cancer?” The response options less than 3 years were coded as “Current” (1), and options greater than 3 years were coded as “Overdue” (0). 

#### 2.2.2. Independent Variables

The key independent variables included self-reported social determinants of health and psychosocial factors. 

Social determinants of health operationalized in this study are economic stability, healthcare access, food security, social and community context, neighborhood and built environment, and education access and quality.

Economic Stability: Two items that constituted economic stability variables were annual income and employment status. The responses were coded as “High” (1) and “Low” (0).Healthcare Access: Two items that constituted health access variables were health insurance coverage and frequency of receiving care in the past 12 months. The responses were coded as “High” (1) and “Low” (0).Food Security: Two items that constituted access to food variables included skipping meals and the ability to afford a balanced diet. The responses were coded as “High” (2), “Medium” (1), and “Low” (0).Social and Community Context: The social context variable (1 item) was discrimination when getting medical care because of race or ethnicity. The responses were coded as “Yes” (1) and “No” (0).Neighborhood and Built Environment: Two items about the environment included access to transportation, medical appointments, work, or getting things needed for daily living, and the residential rural–urban community area zip code. The responses were coded as “Urban” (1) and “Rural” (0).Education Access and Quality: Education access and quality were measured by two items: educational levels and health literacy. The educational levels were categorized into less than high school degree, high school degree, and greater than high school degree. Health literacy measures (3 items) were “knowledge about HPV”, “causes of cervical cancer”, and “knowledge of cervical cancer or HPV vaccine”. The responses were coded as “Yes” (1) and “No” (0).

Psychosocial Variables. Psychosocial measures (5 items) included perceived susceptibility (e.g., worried about getting cancer and beliefs that everything causes cancer), perceived behavioral control (e.g., lack of locus of control over the chances of getting cancer), attitude toward behavior (e.g., beliefs that so many different recommendations about preventing cancer make it hard to know which ones to follow), and perceived severity (e.g., beliefs about the fatality of cancer). The response for the item regarding worry was coded as “Not at all” (0), “Somewhat” (1), and “Agree” (2). The responses for the remaining psychosocial items were coded as “Disagree” (1), “Somewhat” (2), and “Agree” (3). Additional covariates included in our analysis for confounding adjustment included race, age, marital status, cancer type, and the number of people in the household. These covariates were adjusted for in the final model.

### 2.3. Statistical Analysis

We followed HINTS analysis procedures by using the appropriate weight, cluster, and strata variables to obtain weighted population-based estimates of included demographics and prevalence of intention to screen and screening behaviors for US women. Descriptive statistics were calculated for the demographic factors and the covariates. Population-based estimates with 95% confidence intervals (95% CI) were produced for Pap testing behaviors for the overall study population. Bivariate and multivariable binomial logistic regression models were used to assess differences in Pap testing by social determinants of health and psychosocial factors after adjusting for covariates, where adjusted odds ratios (aOR) and 95% CI were estimated. Hierarchical multiple regression analyses were performed to determine the cumulative impact of SODH and psychosocial variables on screening. Analyses were completed using SPSS version 29 and SAS version 9.4 (SAS Institute, Cary, NC, USA) with a priori levels of significance at 0.05 and two-sided hypothesis testing. This study was approved as being exempt from full IRB review by the primary author’s Institutional Review Board due to being considered Non-Human Subject Research.

## 3. Results

### 3.1. Demographic Characteristics

Table 1 below shows the demographic characteristics of the participants and other covariates by study outcomes (i.e., interest in screening and actual screening behaviors). Women (*n* = 2224) with a mean age of 46.96 (±12.53) participated in the study. Overall, 90% of the women reported interest in cervical cancer screening, and the proportion of women with current up-to-date screening was 80.0%. Across the racial/ethnic groups in the study, black/African American (68.51%) and Hispanic (63.94%) populations reported higher interest in cancer screening compared with white ones (57.99%), *p* < 0.001. The distributions of screening behavior across racial/ethnic groups (African American 74.68% vs. Hispanic 79.06% vs. white 74.30%) showed no significant differences, *p* = 0.67. The various age groups (35–49 and 50–64 vs. 21–34) reported screening rates of 81.56%, and 81.14% vs. 75.69% *p* < 0.01, respectively. Women in urban residential areas (77.68%) were more likely to report current screening compared to women who were in rural areas (63.66%).

### 3.2. Interest in Screening

Table 2 shows the SDOH and psychosocial factors as predictors of screening interest and screening behaviors. After adjusting for covariates, respondents who reported low literacy vs. high literacy (aOR: 1.30, 95% CI: 1.04–1.62); low access to food vs. high access to food (aOR: 1.71, 95% CI: 1.10–2.66); low economic stability vs. high economic stability (aOR: 1.37, 95% CI: 1.01–1.86); and lived in rural areas vs. urban areas (aOR: 1.59, 95% CI: 1.26–2.00) were more likely to report high interest in cervical cancer screening.

Women who were not all worried about getting cancer vs. those who were worried about getting cancer (aOR: 10.03, 95% CI: 7.08–14.23), who disagreed that everything caused cancer vs. those who agreed everything caused cancer (aOR: 2.48, 95% CI: 1.82–3.38), and those who disagreed about the fatalism of cancer vs. those who agreed about the fatalism of cancer (aOR: 1.61, 95% CI: 1.15–2.24) were more likely to report high interest in cervical cancer screening.

### 3.3. Screening Behaviors

After adjusting for covariates, women more likely to report being overdue for cervical cancer screening were low literacy vs. high literacy (aOR: 1.62, 95% CI: 1.30–2.02); had limited healthcare access vs. high access to care (aOR: 2.58, 95% CI: 1.58–4.20); had low economic stability vs. high economic stability (aOR: 1.40, 95% CI: 1.05–1.89); lived in rural areas vs. urban areas (aOR: 1.96, 95% CI: 1.55–2.46); had < high school degree vs. >high school (aOR: 1.59, 95% CI: 1.07–2.36) and high school degree vs. >high school (aOR: 1.52, 95% CI: 1.18–1.95). However, respondents with low access to food vs. high access to food (aOR: 0.54, 95% CI: 0.36–0.80) were less likely to be overdue for cervical cancer screening.

In terms of psychosocial factors, women who were not all worried about getting cancer were more likely to be overdue for cervical cancer screening compared to those who were worried about getting cancer (aOR: 1.44, 95% CI: 1.06–1.96). Respondents who disagreed about the fatalism of cancer were less likely to be overdue for screening compared to those who agreed about the fatalism of cancer (aOR: 0.69, 95% CI: 0.49–0.95).

### 3.4. Impacts of SDOH and Psychosocial Factors on Interest in Screening

Table 3 shows the hierarchical multiple regression SDOH and psychosocial factors predicting screening interest. When SDOH factors were added to the model, economic stability and rurality significantly predicted participants’ interest in cervical cancer screening, accounting for 2% of the variance. When the psychological factors were entered into the model, worrying about getting cancer was the only significant predictor of interest in cervical cancer screening, accounting for 10.1% of the variance. The combined effect of SDOH and psychosocial factors on interest in cervical cancer screening was about 12% (Adjusted R^2^ = 0.119)

### 3.5. Impact of SDOH and Psychosocial Factors on Actual Screening Behavior

Table 4 shows the hierarchical multiple regression SDOH and psychosocial factors predicting participants’ actual screening behaviors. When SDOH factors were added to the model, health literacy, access to healthcare, access to food, and rurality significantly predicted participants’ actual cervical cancer screening, accounting for 3% of the variance. When the psychological factors were entered into the model, worrying about getting cancer and fatalism were the significant predictors of participants’ cervical cancer screening behavior, accounting for 1% of the variance. The combined effect of SDOH and psychosocial factors on cervical cancer screening behavior was about 4% (Adjusted R^2^ = 0.039).

## 4. Discussion

This study aimed to examine cervical cancer screening behavior and the role of SDOH and psychosocial factors in a nationally representative sample of women in the United States. In the present study, 90% of respondents expressed a high interest in cervical cancer screening, and 80% reported that they had completed screening. This screening rate is higher than the national average of 72.4%, but still 6.8% below the Healthy People 2030 goal [38]. This finding necessitates the need for continuous public health education to create awareness and public health policies to address some of the social determinant barriers. 

SDOH variables (i.e., low literacy, limited food access, low economic stability, access to healthcare, and rurality) and psychosocial determinants (perceived susceptibility—worrying about getting cancer, and perceived severity—fatalistic beliefs about cancer) demonstrated to be significant predictors of interest in cancer screening and actual screening behaviors, which is consistent with other findings [27,28,29,30]. The SDOH factors accounted for 3% of the variance in screening, indicating that although SDOH factors such as health literacy, access to healthcare, access to food, and rurality impact women’s screening behaviors, the effect is weak among this study population. While the effect (10%) of psychological factors (worrying about getting cancer and fatalism) on women’s interest in screening was moderately strong, the impact (1%) of those factors on actual screening was weaker than SDOH. Future studies should explore why the participants’ high interest did not translate into actual screening behavior. The combined impact of SDOH and psychosocial factors on cervical cancer screening behavior was about 5% (R^2^ = 0.5). These findings indicate that, within this study population, other determinants may explain the screening behaviors of the women. Future studies should expand SDOH and psychosocial variables to further elucidate the extent to which these variables impact screening.

A closer look at the SDOH and psychosocial factors revealed interesting trends. Comparing the screening rates among residents of rural and urban counties, residents of rural counties had lower odds of meeting the USPSTF cervical screening recommendations [31]. A previous study did not find a significant difference in screening rates when stratified by rurality [34], notwithstanding, women residing in rural communities display lower cervical cancer screening rates [39].

The results of our study revealed that economically unstable women have lower odds of meeting the recommended cervical screening protocol. Based on the previous literature, women who were not up-to-date with cervical cancer screening recommendations stated that the cost of screening was the major deterrent [21]. Avoiding the cost of cervical screening is a significant reason why women do not fulfill cervical cancer screening [34]. This finding supports the W.H.O.’s finding that lower socioeconomic status directly influences unfavorable health outcomes [40]. Low economic stability prevents individuals from accessing necessary healthcare [41], consequently, women in this category are less likely to engage in cervical cancer screening behavior. 

To address the disparity in cervical screening among women with low incomes and limited medical insurance, the Center for Disease Control and Prevention established the National Breast and Cervical Cancer Early Detection Program (NBCCEDP). In addition to screening services, the NBCCEDP also caters to the diagnosis and treatment of breast and cervical cancers. Since its inception, the NBCCEDP program has supported 6.2 million women across the United States, with over 5000 cases of invasive cervical cancer and 242,261 cases of premalignant cervical lesions diagnosed [42]. Despite the good intentions of the NBCCEDP program, only 6.8% of eligible women in the United States benefited from the service [42]. Hence, efforts must be channeled to increase awareness and promotion of this service, which will consequently increase cervical cancer screening among women with low incomes and inadequate health insurance. 

Women with low literacy had higher odds of being overdue for cervical cancer screening compared to those with higher literacy in our present study. Women with higher levels of literacy are more up-to-date with cervical cancer screening recommendations and are more likely to have engaged in screening behavior [43].

A combination of factors such as low income and low literacy levels have been noted to result in low patronage of cervical screening [31,44]. High levels of health literacy are needed to generate the right health-related decisions, including cervical cancer screening, among women [45]. However, achieving high health literacy and being well informed are necessary conditions, but not sufficient, to achieve the overall cervical cancer screening goal of 84.3%.

Notably, our findings showed that psychosocial factors, such as perceived cancer risk and fatalism, were significantly associated with cervical cancer screening behaviors. Perceived susceptibility suggests that individuals who believe they are at risk of a disease are more likely to take preventive measures [46]. Conversely, those who perceive they are not susceptible are less inclined to take precautionary measures [46]. We observed that participants who felt they were not susceptible to getting cervical cancer were less likely to consider screening (preventive measure) compared to those who felt they were susceptible to getting cancer. Respondents who agreed that cancer is not always fatal were more likely to reported that they had completed screening compared to those who agreed that cancer is always fatal. Fatalistic beliefs can lead to despondency, negatively influencing individuals’ self-efficacy to accept screening [47]. Thus, the belief that cancer does not always lead to death is a protective factor, and education should emphasize that cervical cancer death can be prevented by routine screening.

In this study, we found that women who had completed cervical cancer screening were between the ages of 34 and 49, namely 81.56%. Women aged 50–64 and 21–34 had lower screening rates of 81.14% and 75.69%, respectively. Previous studies have reported that women between 30 and 65 years old had higher screening rates, compared to younger women [31,48], and this finding is in line with our study. Among women who were up-to-date with cervical cancer screening, 42% were between 31 and 35 years old, while women between 21 and 30 had lower rates of being up-to-date with screening [49]. Beyond the age of 65, cervical cancer screening rates are seen to decline [50]. Possible reasons for lower compliance with screening recommendations in the younger age group may be due to less aggressive public health education to create awareness among this subgroup [48]. Despite the lack of screening in younger women and the lack of awareness, the highest prevalence of cases of Human Papillomavirus (HPV) infection is recorded in teenagers and women in their early 20s [51]. This implies that awareness through intensified education must be addressed to this age group to prevent them from developing invasive carcinoma of the cervix. 

Concerning race, black/African American and Hispanic respondents in our study had a higher prevalence of cervical cancer screening compared to non-Hispanic whites. This finding is consistent with the current trend, with white women having lower rates of cervical cancer screening [34]. A study reported a lack of health insurance as one of the major reasons for white women not undergoing cervical cancer screening [52]. In other study settings, Asian/Pacific Islander women had a higher engagement with screening, followed by Hispanic women, and, lastly, non-Hispanic black women, contrary to this study’s findings [53]. Several interventions have been directed toward increasing cervical cancer screening among minority women [54]. For instance, an intervention study by Jibaja-Weiss resulted in an increase in cervical cancer screening among Mexican women [55]. The cumulative effects of similar interventions targeting minority populations may account for the higher screening rates seen in minority races. 

Despite our findings in this study being valuable to the body of literature on cervical cancer screening, there are potential limitations present within this study. The nature of secondary data analysis may be a limitation of this study. Though the sample is nationally representative and thus has a higher level of generalizability, we still had to remove cases that were inaccurate, incomplete, or missing responses for this data analysis. Secondly, though we gathered important insights on the association SDOH have on screening behaviors, the authors and researchers of this study did not create or refine the questions asked in order to focus solely on this topic of interest. Lastly, the self-reported nature of the HINTS survey may lead to inaccurate responses and potential bias in the results. Though these limitations were present in the study, we do believe the study’s strengths were paramount. Evidence shows that SDOH and psychosocial determinants are associated with cervical cancer screening, but what is unknown is the strength of the associations. To our knowledge, this is the first study that has given a glimpse of the strength of the association. This study’s comprehensive and concurrent evaluation of the two factors (SDOH and psychosocial factors) demonstrates its strength. Using a large sample size is also a strength because it allows us to conduct subgroup analysis. Finally, the strength is that the study provides evidence for the importance of consideration of SDOH within future public health interventions, messaging, and implications to improve cervical cancer screening. 

## 5. Conclusions

The findings of this study underscore that SDOH and psychosocial factors are associated with participants’ interest in screening and their actual screening behaviors. Therefore, well-crafted and intentional public health policies are needed to address social determinants of health (economic stability, health access, food security, social context, and neighborhood and built environment) and psychosocial determinants (perceived susceptibility and severity of getting cancer) known to influence women’s screening behaviors. Using a robust communication model (such as the 3Rs—Reframing, Reforming, and Reprioritizing) to promote HPV self-sampling for cervical cancer screening may help reduce some of the SDOH behavior barriers [56,57]. 

## Figures and Tables

**Table 1 behavsci-14-00811-t001:** Demographic characteristics and other covariates by interest in screening and cervical cancer screening behavior.

	Overall (*n* = 2224)	Interest (*n* = 1844)		Pap Test (*n* = 2224)	
	*n* (%)	No *n* (%)	Yes *n* (%)	Overdue *n* (%)	Current *n* (%)
Race					
Others	164 (7.37%)	43 (31.16%)	95 (68.84) **	51 (31.10%)	113 (68.90%)
Non-Hispanic Black or African American	387 (17.40%)	97 (31.49%)	211 (68.51%) **	98 (25.32%)	289 (74.68%)
Hispanic	468 (21.04%)	141 (36.06%)	250 (63.94%) **	98 (20.94%)	370 (79.06%)
Non-Hispanic Asian	104 (4.68%)	36 (38.71%)	57 (61.29%)	27 (25.96%)	77 (74.04%)
Non-Hispanic White	1101 (49.51%)	384 (42.01%)	530 (57.99%)	283 (25.70%)	818 (74.30%)
Age Range					
21–34 years	469 (21.09%)	177 (43.17%)	233 (56.83%)	114 (24.31%)	355 (75.69%)
35–49 years	244 (10.97%)	81 (38.03%)	132 (61.97%)	45 (18.44%)	199 (81.56%) **
50–64 years	222 (9.98%)	71 (37.97%)	116 (62.03%)	33 (14.86%)	189 (85.14%) **
65+ years	1289 (57.96%)	372 (35.98%)	662 (64.02%)	365 (28.32%)	924 (71.68%)
Marital Status					
Not married	1043(46.90%)	330 (37.97%)	539 (62.03%)	302 (28.95%)	741 (71.05%)
Married	959 (43.12%)	300 (38.17%)	486 (61.83%)	205 (21.38%)	754 (78.62%) **
Live with partner	196 (8.81%)	65 (38.69%)	103 (61.31%)	46 (23.47%)	150 (76.53%)
Residential Area					
Rural ^a^	432 (19.42%)	177 (46.70%)	202 (53.30%)	157 (36.34%)	275 (63.66%)
Urban	1792 (8058%)	524 (35.77%)	941 (64.23%) **	400 (22.32%)	1392 (77.68%) **
Cancer Type					
Gynecological	45 (2.02%)	8 (25.00%)	24 (75.00%)	17 (37.78%)	28 (62.22%)
Breast Cancer	57 (2.56%)	14 (35.90%)	25 (64.10%)	15 (26.32%)	42 (73.68%)
Gastrointestinal	11 (0.49%)	1 (12.50%)	7 (87.50%)	4 (36.36%)	7 (63.64%)
Other	94 (4.23%)	19 (27.54%)	50 (72.46%)	18 (19.15%)	76 (80.85%)
None	1992 (89.57%)	653 (38.96%)	1023 (61.04%)	497 (24.95%)	1495 (75.05%)
Number of People in Household					
One person	525 (23.61%)	170 (38.55%)	271 (61.45%)	155 (29.52%)	370 (70.48%)
Two people	744 (33.45%)	225 (37.13%)	381 (62.87%) *	174 (23.39%)	570 (76.61%) *
Three or more people	955 (42.94%)	306 (38.39%)	491 (61.61%)	228 (23.87%)	727 (76.13%) *

Note: * *p* < 0.01. ** *p* < 0.001. ^a^ County-level rurality was determined using Rural–Urban Continuum Codes (RUCC), which classify US counties from 1 to 9 based on metropolitan size and proximity. Codes 1–3 represent urban counties, while 4–9 denote rural counties, differentiated by urban population size and proximity to metropolitan areas.

**Table 2 behavsci-14-00811-t002:** Social determinants of health and psychosocial factors as predictors of interest in screening and actual cervical cancer screening behaviors (*n* = 2224).

	Interest in Screening			Screening Behavior (Pap Test)		
	AdjOR	95% CI	*p*-Value	AdjOR	95% CI	*p*-Value
Social Determinants of Health						
Health Literacy:						
Low	1.30	1.04–1.62	0.02	1.62	1.30–2.02	0.00
High	1.00			1.00		
Healthcare access:						
Low	0.96	0.58–1.61	0.89	2.58	1.58–4.20	0.00
High	1.00			1.00		
Access to food:						
Low	1.71	1.10–2.66	0.02	0.54	0.36–0.80	0.00
Medium	1.50	0.93–2.24	0.10	0.95	0.62–1.47	0.83
High	1.00			1.00		
Economic Stability						
Low	1.37	1.01–1.86	0.05	1.40	1.05–1.89	0.02
High	1.00			1.00		
Discrimination						
No	1.38	0.98–1.95	0.07	1.02	0.73–1.41	0.93
Yes	1.00			1.00		
Residential Area						
Rural	1.59	1.26–2.00	0.00	1.96	1.55–2.46	0.00
Urban	1.00			1.00		
Education						
<High School Degree	0.78	0.50–1.22	0.28	1.59	1.07–2.36	0.02
High School Degree	1.12	0.87–1.46	0.38	1.52	1.18–1.95	0.00
>High School Degree	1.00			1.00		
Psychosocial Factors						
Worry about getting cancer						
Not at all	10.03	7.08–14.23	0.00	1.44	1.06–1.96	0.02
Somewhat	2.70	2.10–3.47	0.00	1.09	0.86–1.38	0.47
Agree	1.00			1.00		
Everything caused cancer						
Disagree	2.48	1.82–3.38	0.00	0.96	0.70–1.32	0.79
Somewhat Agree	1.66	1.30–2.12	0.00	0.98	0.77–1.26	0.89
Agree	1.00			1.00		
Impossible to prevent cancer						
Disagree	0.93	0.62–1.38	0.70	0.72	0.50–1.06	0.10
Somewhat	0.86	0.60–1.24	0.42	0.68	0.48–0.97	0.03
Agree	1.00			1.00		
Many recommendations cause confusion.						
Disagree	1.28	0.88–1.87	0.20	0.89	0.61–1.30	0.55
Somewhat	0.98	0.77–1.25	0.89	0.89	0.70–1.13	0.34
Agree	1.00			1.00		
Fatalism						
Disagree	1.61	1.15–2.24	0.00	0.69	0.49–0.95	0.02
Somewhat	1.26	0.99–1.60	0.06	0.78	0.62–0.99	0.04
Agree	1.00			1.00		

Abbreviations: AdjOR = adjusted odds ratio; CI = confidence interval. Covariates were adjusted for.

**Table 3 behavsci-14-00811-t003:** Social determinants of health and psychosocial factors as predictors of women’s interest in cancer screening behaviors (*n* = 2224).

	Model 1			Model 2			Model 3		
Variable	B	SE B	β	B	SE B	β	B	SE B	β
Covariates									
Age	0.01	0.01	0.03	0.02	0.01	0.05	0.03	0.01	0.07 **
Family History	0.02	0.03	0.02	0.03	0.03	0.03	0.01	0.02	0.01
Diagnosed	0.07	0.10	0.03	0.10	0.10	0.04	0.01	0.10	0.00
Cancer Type	0.02	0.03	0.02	0.02	0.03	0.02	−0.01	0.03	−0.01
Marital Status	0.00	0.02	0.01	0.00	0.02	0.00	−0.01	0.02	−0.01
Number of Household	0.01	0.02	0.01	0.00	0.02	0.00	0.00	0.02	0.00
Race	−0.03	0.01	−0.09	−0.03	0.01	−0.09	−0.05	0.01	−0.13 **
Social Determinants									
Health Literacy				0.06	0.03	0.06	0.03	0.03	0.03
Access to Healthcare				0.00	0.07	0.00	−0.01	0.07	0.00
Access to Food				0.05	0.02	0.06	0.04	0.02	0.04
Economic Stability				0.10	0.04	0.07 *	0.09	0.04	0.06 *
Residential Area				0.11	0.03	0.09 *	0.09	0.03	0.07 **
Education				−0.04	0.03	−0.04	−0.02	0.02	−0.03
Psychosocial Factors									
Worried							0.24	0.02	0.32 **
Cause cancer							0.03	0.02	0.03
Impossible to prevent							−0.04	0.02	−0.04
Recommendation							0.02	0.03	0.03
Fatalism							−0.02	0.02	−0.03
Adjusted R^2^	0.005			0.020			0.119		
*F* for change in R^2^	0.010 **			0.019 **			0.107 **		

Note:* *p* < 0.01. ** *p* < 0.001. Overall model: *F*(18, 1492) = 12.32, *p* < 0.001. Dependent variable is interest in cancer screening; B = unstandardized coefficient; SE B = standard error of the coefficient; β = standardized coefficient.

**Table 4 behavsci-14-00811-t004:** Social determinants of health and psychosocial factors as predictors of women’s cervical cancer screening behaviors (*n* = 2224).

	Model 1			Model 2			Model 3		
Variable	B	SE B	β	B	SE B	β	B	SE B	β
Covariates									
Age	−0.02	0.01	−0.05	−0.02	0.01	−0.04	−0.01	0.01	−0.04
Family History	−0.03	0.02	−0.04	−0.02	0.02	−0.03	−0.03	0.02	−0.03
Diagnosed	0.16	0.08	0.07	0.15	0.08	0.07	0.14	0.08	0.06
Cancer Type	0.04	0.03	0.05	0.03	0.03	0.04	0.03	0.03	0.03
Marital Status	0.04	0.02	0.06	0.04	0.02	0.06	0.03	0.02	0.05
Number of Household	0.01	0.01	0.02	0.01	0.01	0.02	0.01	0.01	0.02
Race	−0.01	0.01	−0.02	−0.01	0.01	−0.03	−0.01	0.01	−0.04
Social Determinants									
Health Literacy				0.08	0.02	0.08	0.08	0.02	0.08 **
Access to Healthcare				0.14	0.06	0.06	0.14	0.06	0.05 *
Access to Food				−0.04	0.02	−0.06	−0.05	0.02	−0.06 *
Economic Stability				0.04	0.03	0.03	0.04	0.03	0.03
Residential Area				0.11	0.03	0.10	0.10	0.03	0.10 **
Education				0.01	0.02	0.01	0.01	0.02	0.02
Psychosocial Factors									
Worried							0.06	0.02	0.08 **
Cause cancer							−0.01	0.02	−0.02
Impossible to prevent							0.03	0.02	0.03
Recommendation							−0.01	0.02	−0.01
Fatalism							−0.04	0.02	−0.05 *
Adjusted R^2^	0.009			0.034			0.039		
*F* for change in R^2^	0.013 **			0.029 **			0.007 **		

Note:* *p* < 0.01. ** *p* < 0.001. Overall model: *F*(18, 1774) = 5.05, *p* < 0.001. The dependent variable is cervical cancer screening behavior (pap test); B = unstandardized coefficient; SE B = standard error of the coefficient; β = standardized coefficient.

## Data Availability

The data presented in this study are publicly available data at https://hints.cancer.gov/data/download-data.aspx (accessed on 9 January 2024).

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
