# Peer review of "Exploring Cervical Cancer Screening Uptake among Women in the United States: Impact of Social Determinants of Health and Psychosocial Determinants"

_behavsci, 2024, doi:10.3390/bs14090811_

Round 1

Reviewer 1 Report

Comments and Suggestions for Authors

The paper makes a useful contribution to the literature on the determinants of uptake. However, the paper is unclear in several places and I have several suggestions to improve the paper:

The authors refer to 'social determinants of health' as a theory behind their study. However, is something related to the social determinants of health behaviour more appropriate as uptake of screening is a health behaviour rather than health itself? The authors may need to be more explicit on the difference, if relevant

It would be of benefit to international readers to understand how a screening is organised and in what setting (primary care?)? Are eligible women contacted by the family practitioner? This would also help provide the reader with an understanding of what factors may determine uptake. 

Related to the previous point there is some literature (both quant and qual) with a non-US focus that considers factors related to uptake i.e Bang et al 2012 JPH, Urwin et al 2023 JPH, Marlow et al 2019 BMC Women's Health. They find healthcare supply characteristic have an influence on uptake.

p2 l82: 'cumulative' seems to be a strange word and implies some interaction (and is used in other sections). Do the author's mean effect of these variables in the same model, i.e the impact of SDOH variables holding psychosocial factors constant and vice versa? 

p3 l108: do the authors have any area-level information on the respondent, for example, if they reside in an area with a higher number of women of screening age or with fewer doctors able to provide a screening? 

p4 l123: do the authors have the number of children in the household and if someone has an informal caregiving responsibility? These may impact on an individual's time and thus whether they go for screening

p4 l124: do the authors not mean these variables were used to adjust for the other independent variables? Or confounding factors?

p4 l148: I am not clear what this means - apply how? Do the authors mean they apply sample weights to regression analyses to ensure results are nationally representative?

p4 l152: bivariate instead of binary? or does binary refer to the outcome variable?

Table 1: add years to age variable label. How is rural and urban defined?

Table 2: For regression analysis did the authors not consider if interest in screening and having had a screening were related? Presumably interest in screening may be affected by a bad screening experience as the qualitative literature shows. 

Table 2: How are categorical variables included? For example age is only one row but in the summary statistics this had several categories? The same applies for other variables

Table 3: recommend using adjusted r squared instead because this accounts for the different number of variables in each model 

Reviewer 2 Report

Comments and Suggestions for Authors

This was the study of behavior science. The detail of table should be corrected to remove redundant word. There was double header of model 3 in Table 4. The sentences were too long and difficult to read. Simplified sentences were needed.

Comments on the Quality of English Language

This was the study of behavior science. The detail of table should be corrected to remove redundant word. There was double header of model 3 in Table 4. The sentences were too long and difficult to read. Simplified sentences were needed.

Reviewer 3 Report

Comments and Suggestions for Authors

This is a very interesting study investigating cervical cancer screening in North American Women, and the effect of social determinants of health on screening. The paper is well-written and of interest for the journal; However, several minor changes are recommended before considering it for publication.

ABSTRACT.

A couple of lines introducing the topic of research would be recommended, before the description of the objectives of the paper. 

In the methods section of the abstract, I recommend to add, shortly, some variables that were recorded in the retrospective cross-sectional study.

The role of public health should be emphasized in the abstract section.

INTRODUCTION

Before starting the introduction about cervical cancer, I recommend to start with a brief synthesis of some of the most common gynecological cancers in women in the United States. Furthermore, I recommend to mention which kind of screenings are recommended to women with regard to gynecological cancers ( e.g. breast cancer, cervical cancer, etc). 

Are the hypothesized social determinants of health specific for cervical cancer or for other gynecological cancers? What about nicotine consumption?

Life-style medicine should be also introduced in the introduction section of the paper.

I recommend to clarify the differences between social determinants of health and psychosocial determinants. The authors use both terms in the whole paper.

MATERIAL AND METHODS

A short table presenting the main outcome variables and independent variables would be helpful for the readers.

What does this sentence means? Please, clarify or rephrase it: "This study was approved as exempt by...".

RESULTS

A figure presenting the factors influencing the screening of cervical cancer would be helpful. Table 3 is really extensive and detailed for results of the statistical analyses. However, a figure representing the weight of each factors is necessary in my point of view.

DISCUSSION

In line 329 the authors discuss one of their findings. They described that Black/African American and Hispanic respondents had a higher prevalence of cervical cancer cmpared to non-Hispanic whites. Does it mean that native American respondents showed lower rates of cervical cancer screening? This sentence should be clarified. These findings are in contrast with the previous literature in the field. Can the authors hypothesize the reasons?

CONCLUSIONS

Specific campaign proposals can be recommended to fill this gap in screening. I recommend the authors to increase details of sich proposals.
